# Prevalence of Pathogens and Other Microorganisms in Premenopausal and Postmenopausal Women with Vulvovaginal Symptoms: A Retrospective Study in a Single Institute in South Korea

**DOI:** 10.3390/medicina57060577

**Published:** 2021-06-04

**Authors:** Jong-Chul Baek, Hyen-Chul Jo, Seon-Mi Lee, Ji-Eun Park, In-Ae Cho, Joo-Hyun Sung

**Affiliations:** 1Department of Obstetrics and Gynecology, College of Medicine, Gyeongsang National University, Gyeongsang National University Changwon Hospital, 11, Samjeongja-ro, Seongsan-gu, Changwon-si 51472, Gyeongsangnam-do, Korea; gmfather@gnuh.co.kr (J.-C.B.); tjsal4142@gnuh.co.kr (S.-M.L.); jl1104@hanmail.net (J.-E.P.); 2Department of Obstetrics and Gynecology, College of Medicine, Gyeongsang National University, Gyeongsang National University Hospital, Jinju 52727, Gyeongsangnam-do, Korea; dew8274@hanmail.net; 3Department of Occupational and Environmental Medicine, Institute of Health Sciences, College of Medicine, Gyeongsang National University, Gyeongsang National University Changwon Hospital, Jinju 51472, Gyeongsangnam-do, Korea; yadaf@gnuh.co.kr

**Keywords:** vulvovaginal symptoms, multiplex real-time PCR test, *Ureaplasma parvum*, menopause, *Gardnerella vaginalis*

## Abstract

*Background and Objectives:* Vaginitis causes vulvovaginal symptoms, including itching, irritation, vaginal discharge, burning, or foul odor. It is one of the most common diseases encountered in gynecologic practice. Hypoestrogenism due to menopause has a considerable negative effect on vaginal health and leads to changes in the vaginal pH and vaginal microflora, which are related to a change in the causes and microorganisms of vaginitis. Thus the aim here was to investigate the prevalence of pathogens and other microorganisms in premenopausal and postmenopausal women with vulvovaginal symptoms, using an STD 12-Multiplex real-time PCR test and routine culture of vaginal discharge. *Materials and Methods:* From January 2018 to December 2019, records of patients diagnosed with vaginitis at Changwon Gyeongsang National University Hospital were retrospectively reviewed. The premenopausal and postmenopausal subjects were categorized into Group A and Group B, respectively. and the data of symptoms, general characteristics, and results of both STD 12-Multiplex real-time PCR test and routine culture of vaginal discharge were retrieved. *Results:* On the STD 12-Multiplex real-time PCR test, *Gardnerella vaginalis* was the most common microorganism in both groups. *Ureaplasma parvum* was the second most common one, followed by *Candida* speceies. On the routine culture of vaginal discharge, *Escherichia coli* was the most common aerobic bacterial microorganism in both groups, followed by *Streptococcus agalactiae* (Group B Streptococcus, GBS). There was no significant difference between the two groups. Pathogens and other microorganisms of patients with vulvovaginal symptoms that showed a statistically significant difference between the two groups were *Ureaplasa parvum*, *Ureaplasma urealyticulum*, *Trichomonas vaginalis*, and *Staphylococcus aureus*. *Conclusion:* In this study, the prevalence of pathogens and other microorganisms in menopausal women with vulvovaginal symptoms did not show a significant difference from premenopausal women. Therefore, management strategies for patients with vulvovaginal symptoms should be developed through accurate diagnosis using appropriate diagnostic methods.

## 1. Introduction

Vaginitis causes vulvovaginal symptoms, including itching, irritation, vaginal discharge, burning, or foul odor. It is one of the most common diseases encountered in gynecologic practice. Patients with vulvovaginal symptoms should be distinguished according to noninflammatory vaginitis and inflammatory vaginitis. Common causes of vaginitis are bacterial vaginosis, trichomoniasis, and vulvovaginal candidiasis. Among these, bacterial vaginosis is the most common noninflammatory vaginitis, and the prevalence of bacterial vaginosis is reported to range from 20% to 60% [1]. Most symptoms of vaginitis are not specific to the cause, and the common symptoms are pruritus, burning sensation, irritation, discharge, and perineal discomfort. It is difficult to diagnose the cause of vaginitis based on symptoms alone. Since various pathogens are associated with vaginitis, the diagnostic methods vary depending on the causes. Gram staining and Nugent scoring, Amsel’s criteria using saline microscopy, molecular diagnostic assays, and nucleic acid amplification tests have been suggested for diagnosing bacterial vaginosis. Among these, several tests using polymerase chain reaction (PCR) have been performed in recent years, and the accuracy of diagnosis with such tests did not show a significant difference compared to that with Gram stain with Nugent scoring, which is the standard method. The risk factors for vaginitis also vary according to the pathogen. Among these, hypoestrogenism due to menopause has a considerable negative effect on vaginal health and leads to changes in the vaginal pH and vaginal microflora, which are related to a change in the causes and microorganisms of vaginitis. The lack of appropriate diagnosis and treatment of vaginitis might increase the risk of preterm labor pain, incidence of premature rupture of membranes (PROM), and amniotic fluid infection in pregnant women [2,3], as well as cause pelvic inflammatory diseases such as cervicitis, endometritis, salpingitis, and sexually transmitted infections, including human immunodeficiency virus (HIV) [4,5].

The purpose of this study was to determine the prevalence of pathogens and other microorganisms in premenopausal and postmenopausal women with vulvovaginal symptoms, using STD 12-Multiplex real-time PCR test and routine culture of vaginal discharge.

## 2. Materials and Methods

From January 2018 to December 2019, records of patients diagnosed with vaginitis at Changwon Gyeongsang National University Hospital were retrospectively reviewed, and the data of symptoms, general characteristics, and results of both STD 12-Multiplex real-time PCR test and routine culture of vaginal discharge were retrieved. The total number of patients diagnosed with vaginitis during this period was 528 patients. Among these, the number of excluded patients in this study was 189 patients. Exclusion criteria were: patients who had undergone a hysterectomy (*N* = 31), pregnant patients (*N* = 113), patients with incomplete medical records (*N* = 18), and patients who received treatment for vaginitis within 30 days before evaluation (via oral or vaginal) (*N* = 27). A total of 339 patients were included in the study. The premenopausal and premenopausal subjects were categorized into group A and group B, respectively. There were 157 and 182 patients in groups A and B, respectively. Postmenopausal state was defined as amenorrhea for more than 12 months in women aged 50 years or more, or the diagnosis of menopause with a follicle stimulating hormone (FSH) level of 25 mIU/mL or more in women aged 50 years or younger. Each patient’s age, parity, body mass index (BMI), smoking status, current use of IUD (intrauterine device), underlying disease conditions, use of menopause hormonal therapy (MHT), and symptoms were investigated. Two vaginal samples were obtained from each patient. One sample for routine culture was collected by swab from the vaginal posterior fornix to the lower portion of the vagina, and the other sample for STD 12-Multiplex real-time PCR test was collected by swab cervix and vaginal posterior fornix. The collected vaginal swab samples were transported to the laboratory in the transport medium. AM608-1S or AM608-2S for both routine culture and the STD 12-Multiplex real-time PCR test. STD12-Multiplex real-time PCR test detected *Gadnerella vaginalis*, *Candida albicans*, *Mycoplasma hominis*, *Mycoplasma genitalium*, *Ureaplasma parvum*, *Ureaplasma urealyticulum*, *Chlamydia trachomatis*, *Trichomonas vaginalis*, Herpes simplex virus type 2, *Treponema pallidum*, and *Neisseria gonorrheae.* Aerobic bacteria and *Candida* spp. were detected on culture. STD12-Multiplex real-time PCR test that extracts the genomic DNA from the vaginal swab specimen was performed by using an automated DNA extraction instrument (ExiPrepTM 16 Dx, Bioneer Co., Daejeon, Korea) and the ExiPrepTM Dx bacteria genomic DNA kit (Bioneer Co.). AccuPower STI 8A, 8B, 4C-plex real-time PCR kit (Bioneer Co.) was used as a premix kit. To prepare the PCR mixture, diethyl pyrocarbonate-treated distilled water (44 µL), internal positive control (1 µL), and extracted nucleic acid (5 µL) were added to the PCR premix tubes according to the manufacturer’s instructions. Before the PCR reaction, the tube contents were mixed using ExiSpinTM (Bioneer Co.) to dissolve the premix pellet completely. PCR was conducted using an ExicyclerTM 96 (Bioneer Co.). Among the result of bacteria tested on routine culture, the *Streptococcus anginosus* group includes *Streptococcus anginosus*, *Streptococcus intermedius*, and *Streptococcus constellatus.* The *Corynebacterium* group includes *Corynebacterium striatum*, *Corynebacterium simulans*, and *Corynebacterium amycolatum*. Meanwhile, the other groups, which are smaller in number, include *pantoea* species, *Citrobacter freundii*, and *proteus mirabillis*. Sexually transmitted diseases (STDs) included *Chlamyida trachomatis*, *Mycoplasma genitalium*, *Trichomonas vaginalis*, and Herpes simplex virus type 2 (HSV-2).

Categorical variables were expressed as frequency (n) with percentage (%), and continuous variables were expressed as means (M) with standard deviations (SD). Continuous variables were compared using Student t-tests. Categorical variables were compared using Chi-square tests in the case of a theoretical chi-squared distribution, and Fisher’s exact tests in the case of an expected cell count less than 5. Statistical analysis was performed using IBM SPSS Statistics for Windows, version 24.0 (IBM Inc., Armonk, NY, USA), and a *p*-value of less than 0.05 was considered statistically significant.

All procedures carried out in this study involving human participants needed ethical approval were in accordance with the Declaration of Helsinki in 1964 and subsequently amended or similar ethical standards. This study was approved the Institutional Review Board of Gyeongsang National University Changwon Hospital (IRB file No. 2021-03-010). All methods were performed in accordance with the relevant guidelines and regulations of institution.

## 3. Results

Table 1 shows general characteristics of all patients in this study. Groups A and B included 157 and 182 patients, respectively. Among the underlying diseases, patients with hypertension, diabetes mellitus, hyperlipidemia, and cerebrovascular disease were more common in Group B. However, the prevalence of thyroid disease and breast cancer did not show any difference between the groups. Moreover, fifteen and two patients in groups A and B respectively were using IUD. In group B, 11 patients received menopause hormone therapy.

Among vulvovaginal symptoms, vaginal discharge was the most common complaint in both groups, reported by 112 patients (71.3%) in group A and 102 (56%) in group B. Most patients in both groups had only one symptom, and three patients in group A and 6 in group B complained of three symptoms. Among patients with two symptoms, vagina discharge with lower abdominal pain was the most common symptom (*n* = 21), followed by vaginal discharge with itching sensation (*n* = 17), vaginal discharge with foul odor (*n* = 12), and vaginal discharge with spotting (*n* = 10).

Table 2 shows the various pathogens and other microorganisms as identified using both tests. On the STD 12-Multiplex real-time PCR test, *G. vaginalis* was the most common microorganism in both groups, seen in 104 patients in group A and 116 patients in group B. *U. parvum* was the second most common microorganism detected in 81 patients in group A and 49 in group B, followed by Candida species, seen in 31 and 29 patients in groups A and B respectively. Moreover, 152 patients were detected with aerobic bacterial microorganisms through a routine culture test of vaginal discharge, including 62 (39.5%) and 90 (49.4%) patients in groups A and B, respectively, and there was no statistically significant difference between the two groups. A further 70 patients were detected with Candida species through both tests, including 32 and 38 patients in group A and group B, respectively. Among five patients with non-albicans Candida in group B, two patients were co-infection with C. albicans. There was no statistically significant difference in either group. In this study, the prevalence of Candida species was 70 (20.6%). *Escherichia coli (E. coli)* was the most common aerobic bacterial microorganisms in both groups, followed by *Streptococcus agalactiae* (Group B Streptococcus, GBS). The other bacterial microorganisms showed slight differences in sequence between two groups. In single detection with aerobic bacterial group, the most common bacteria were *E. coli* (*n* = 15), followed by GBS (*n* = 7), *Enterococcus faecalis* (*E. faecalis*) (*n* = 6), *Klibsiella pneumoniae* (*K. pneumonia*) (*n* = 2), *Staphylococcus aureus* (*S. aureus*) (*n* = 2), and *Pseudomonas aeruginosa* (*P. aeruginosa*) (*n* = 1). Only the prevalence of *S. aureus* showed a statistically significant difference.

Sexually transmitted disease (STD) was seen in 12 and nine patients in Group A and Group B, respectively. After excluding HSV-2 from the STD pathogens, there were five in group B (four patients of *Trichomonas vaginalis*, one patient of *Chlamydia trichomatis*) and three patients (two with *Chlamydia trachomatis*, one with *M. genitalium*) in group A who had STD. Eight patients in group A, of which one patient had a concomitant infection with trichomonas, and six in group B were positive for herpes simplex type 2.

In addition, 11 patients in group A and 19 in group B showed no causative pathogens and other microorganisms for vulvovaginal symptoms on either test. There was no statistically significant difference between Group A and Group B.

Pathogens and other microorganisms as identified by both tests that showed a statistically significant difference between the two groups were *U. parvum*, *U. urealyticulum*, *Trichomonas vaginalis*, and *S. aureus*.

## 4. Discussion

This study investigated the prevalence of pathogens and other microorganisms in premenopausal and postmenopausal women with vulvovaginal symptoms using STD 12-Multiplex real-time PCR and routine culture of vaginal specimens.

Our study showed that 220 patients (64.9%) were diagnosed with *G. vaginalis* on PCR, including 104 (66.2%) and 116 (63.7%) patients in groups A and B, respectively. It was the most common microorganism of vaginitis in both groups. Bacterial vaginosis is the most common cause of vaginitis, accounting for 40–50% of the cases in which the cause is identified [6]. One study using Gram stain and Nugent scoring system showed that the prevalence of bacterial vaginosis was 40.5% [7]. The prevalence of *G. vaginalis* on PCR was 64.9% in this study. The method used in this study was to detect only *G. vaginalis* in the diagnosis of bacterial vaginosis. It is difficult to determine the prevalence of bacterial vaginosis by testing only for *G. vaginalis* because 98–100% of women with and 55% without bacterial vaginosis have *G. vaginalis.* In spite of this lack of specificity, *G. vaginalis* is still an important bacterium in the causes of bacterial vaginosis. Although many methods have been proposed, Gram staining and Nugent scoring are the standard methods for diagnosis [6]. In recent times, molecular tests have been introduced, and the results of DNA probes for *G. vaginalis* have a sensitivity and specificity of 92–100% and 92–98%, respectively, compared to Gram stain [8]. These tests have also shown higher sensitivity than the original Amsel’s test (92.7% vs. 75.6%; *p* < 0.0001) [9]. The recent results of a real-time multiplex PCR assay for bacterial vaginosis have a sensitivity of 91.7% and a specificity of 86.6%, compared to Nugent score and/or Amsel’s criteria [10]. A previous study reported that sexually active women had a higher prevalence of bacterial vaginosis than the menopausal group, which is different from the results of this study [11]. Menopause increases the incidence of vaginitis due to several pathophysiological mechanisms. During menopause, vaginal tissues become thinner and lose elasticity due to the reduced estrogen and progesterone levels [12]. Changes in the vaginal pH, vaginal microflora, and cellular glycogen content occur due to endocrine changes caused by menopause [13]. Menopausal women showed a more alkaline vaginal condition (pH > 4.5) [14]. In addition, asymptomatic carriers of *G. vaginalis* and gram-negative bacteria are more common among postmenopausal women [15,16]. These factors play an important role in the development of vulvovaginal symptoms and vaginal infection by inducing an imbalance in the vaginal microflora. Being sexually active is one of the many risk factors associated with vaginitis such as bacterial vaginosis, and menopause might also be included in this group.

Candida species were the third most common those identified in this study. In this study, both PCR test for *C. albicans* and culture were used for the diagnosis of vulvovaginal candidiasis (VVC). The sensitivity of office microscopy, which is used as the first line for VVC diagnosis, is only about approximately 50% [17]. Thus, a yeast culture is required for confirmation of the diagnosis. The sensitivity and specificity of the methods using an antigen or DNA probe are 79–97% and 96–99%, respectively, compared to the method of diagnosis using culture of vaginal discharge [18,19]. Although VVC cannot be diagnosed as a detection of Candida species by culture test, positive result of those in patients with vulvovaginal symptoms plays role in the diagnosis of VVC. *C. albicans* is the most common cause of 85% of VVC [20]. This study presents that overall was 85.7%, group A was 90.6%, and group B 81.6%. Organisms other than *C. albicans*, which accounted for 30%, were related to chronic or recurrent disease [21]. These were *Candida glabrata*, *Candida tropicalis*, and *Candida krusei* (seven, two, and one, respectively). Among these, *C. glabrata* is known to occur more frequently in patients with diabetes. In this study, five out of seven patients of *C. glabrata* were diagnosed with diabetes. The correlation could not be evaluated because the number of patients was small.

In this study, the aerobic bacterial microorganisms were E. coli, Streptococcus agalactiae (GBS), Enterococcus faecalis, S. aureus, Klebsiella pneumonia, Streptococcus anginosus group, Corynebacterium group, Enterobacter aerogenes, and Pseudomonas aeruginosa in descending order. Compared to other studies [22], the order was different, but the causative bacteria were similar. The culture test of the vaginal discharge used in this study can be misleading and indicate normal vaginal flora such as E. coli, GBS, and enterococci as pathogens. Therefore, when only culture is used to diagnose aerobic vaginitis and causes of vaginitis as this study, attention must be taken in its interpretation.

No causative organisms for vulvovaginal symptoms were found in 11 (7.0%) and 19 (10.4%) patients in group A and group B, respectively. The prevalence of non-infectious and inflammatory vaginitis is less than other types of vaginitis, accounting for about 5–10% of all cases of vaginitis [23]. The causes include atrophic, irritant, allergic, or physiologic discharge.

In this study, *U. parvum* was the second most commonly detected species after *G. vaginalis*. The overall prevalence of *U. parvum*, *U. urealyticulum*, *M. hominis*, and *M. genitalium* was 38.3%, 10.6%, 10.3%, and 0.3%, respectively. *U. parvum* and *U. urealyticulum* were more common in Group B. Many studies about the prevalence of genital sexually transmitted infections (STIs), such as *mycoplasma* and *ureaplasma*, have been published. In a study involving 799 healthy women, the prevalence of *U. parvum*, *U. urealyticum*, *M. hominis*, and *M. genitalium* was 42.7%, 7.6%, 9.9%, and 1.0%, respectively. The prevalence of these organisms was 21.0%, 8.1%, 10.1%, and 0% in those aged above 50 years, which is similar to group A (26.9%, 7.1%, 11.5%, and 0.5%) in our study, and 50.1%, 7.2%, 9.6%, and 1.97% in those aged below 50 years, which is similar to group B (51.6%, 14.6%, 8.9%, 0%, respectively) [24]. There was no significant difference in the results of prevalence. In other studies, the prevalence of *U. parvum*, *U. urealyticulum*, *M. hominis*, and *M. genitalium* was 38.3%, 9.0%, 8.6%, and 0.5%, respectively [25]. These results also showed no significant difference from those of our study. It is difficult to differentiate the pathogenic effect of *U. parvum* between patients with vaginitis from previously described studies and our study. First, although the study by Kim et a. were conducted on healthy women in the same country as ours, there was no difference in the prevalence of *U. parvum* between our study and that study. Second, the prevalence of *U. parvum* was not different between the symptomatic and asymptomatic groups in the previous study (31.8% vs. 32.4%) [25]. Additionally, only 8.5% (11/130 patients) of *U. parvum* was a single infection of *U. parvum* in our study. Some studies have argued that symptoms are caused by serotype. Among serotypes 1, 3/14, and 6, only serovar 3/14 had pathologic effects that caused symptoms [26]. During pregnancy, serovar 3 colonization in the vagina is associated with adverse outcomes, such as spontaneous preterm birth and extremely low gestational age. This relationship has not been observed in serovars 1 and 6 [27]. More studies are necessary to understand the clinical role of *U. parvum* as a causative agent for STI or vaginitis. There has been controversy regarding treatment of *M. hominis* in patients with vulvovaginal symptoms. In our study, *M. hominis* was detected in 35 patients. Among patients with *M. hominis*, *G. vaginalis* was detected in 34 patients, and one patient also had *U. urealyticulum*. In one study, *M. hominis* was not found in patients without *B. vaginosis* [28]. A single *M. hominis* infection did not cause vaginitis or cervicitis. These were not caused by *M. hominis*, but were caused by other pathogens, especially *B. vaginosis*.

Trichomoniasis, which is a sexually transmitted disease (STD), is a common cause of vaginitis along with bacterial vaginosis and vulvovaginal candidiasis. It accounts for approximately 15–20% of all vaginitis infections. It is diagnosed by observing motile and flagellated protozoa using saline microscopy. The culture of vaginal specimens has been regarded as the golden standard for the diagnosis of *T. vaginalis*. There are disadvantages of low sensitivity of microscopy and a long time to obtain the results of the culture method. In recent years, a molecular test that shows high sensitivity compared to method of microscopy is recommended [29]. Only about 50% of women infected with *T. vaginalis* have symptoms such as itching, a foamy vaginal discharge, and dyspareunia [30]. In this study, 4 patients (1%) with trichomoniasis were found only in Group A. All patients complained of vaginal discharge, and 3 patients of those complained of both vaginal discharge and itching, and 1 patient complained of both vaginal discharge and burning. The prevalence found herein was lower than that expected in practice. The prevalence of *T. vaginalis* infection varies from region to region, and annual incidence of South Korea is reported to be about 537.2 patients per 100,000 female persons. The age group with highest incidence was 40–49 years [31]. No case was detected in other studies using microscopy [7] and another report stated that vaginitis of *T. vaginalis* was uncommon in less than 1% of cases [29]. This may be because *T. vaginalis* shows a difference in incidence rate from region to region, and may also vary according to research institutes. It was expected that the prevalence of STD would be higher in the reproductive age, a sexually active group. However, different results were obtained in the present study. In this study, the number of patients diagnosed with STD in this study was small, and it is difficult to confirm these data. *Neisseria gonorrhoeae* was not detected in this study.

This study has several limitations. Since it was conducted in a single institution, which is a tertiary care center, there might be a selection bias. Second, the results of PCR and routine culture were not compared with reference methods such as Gram staining. Third, since this study was performed retrospectively, the past medical history and medication details of patients were incomplete. Fourth, our study did not investigate desquamative inflammatory vaginitis (DIV) and cytolytic vaginosis, which may be other causes of vaginitis. However, this study has the advantage of identifying prevalence of various pathogens and other microorganisms causing vulvovaginal symptoms, including *mycoplasma* and *ureaplasma*, by performing both PCR tests and routine culture in all patients diagnosed with vaginitis. In addition, although the number of samples in this study was small, it can be said that the number of those is sufficient because all patients in this study were limited to cases diagnosed with vaginitis.

## 5. Conclusions

Menopause induced many changes in the vaginal environment such as vaginal microflora and vaginal pH, which play an important role in the occurrence of vulvovaginal symptoms. In this study, the prevalence of pathogens and other microorganisms in menopausal women with vulvovaginal symptoms did not show a significant difference from premenopausal women. Management strategies for patients with vulvovaginal symptoms should be developed through accurate diagnoses using appropriate diagnostic methods.

## Figures and Tables

**Table 1 medicina-57-00577-t001:** General characteristics of patients.

Variable	Group A (*n* = 182)	Group B (*n* = 157)	*p*-Value
Age	41.7 ± 6.5	63.4 ± 11.6	<0.001 *
Parity	1.5 ± 0.9	2.3 ± 1.3	<0.001 *
Duration of menopause		13.0 ± 11.9	
BMI	22.8 ± 3.3	23.5 ± 3.5	0.084 *
Smoking	9 (5.7)	8 (4.4)	0.574 †
IUD	15 (9.6)	2 (1.1)	<0.001 †
Underlying conditions			
Hypertension	8 (5.1)	64 (35.2)	<0.001 †
Diabetes mellitus	4 (2.5)	25 (13.7)	<0.001 †
Thyroid disease			
Hypothyroidism	4(2.5)	10 (5.5)	0.174 †
Hyperthyroidism	2 (1.3)	1 (0.5)	0.598 ‡
Thyroid cancer	4 (2.5)	5 (2.7)	1.000 ‡
Hyperlipidemia	1 (0.6)	15 (8.2)	0.001 †
Breast cancer	4 (2.5)	3 (1.6)	0.708 ‡
Cerebrovascular disease	0 (0)	15 (8.2)	<0.001 †
Hysterectomy	5 (3.2)	26 (14.3)	<0.001 †
MHT	0 (0)	11 (6.0)	0.002 †
Symptom			
Vaginal discharge	112 (71.3)	102 (56.0)	0.004 †
Itching (pruritis)	25 (15.9)	24 (13.2)	0.475 †
Vaginal spotting	14 (8.9)	21 (11.5)	0.429 †
Burning sensation	10 (6.4)	21 (11.5)	0.100 †
Lower abdominal pain	25 (15.9)	17 (9.3)	0.067 †
Skin erosion	12 (7.6)	15 (8.2)	0.839 †
Perineal discomfort	9 (5.7)	13 (7.1)	0.599 †
Foul odor	11 (7.0)	10 (5.5)	0.565 †
Vesicle or ulceration	4 (2.5)	5 (2.7)	1.000 ‡
Urinary symptoms	2 (1.3)	4 (2.2)	0.690 ‡
Number of Symptom			
One	93 (59.2)	138 (75.8)	0.001 ‡
Two	61 (38.9)	38 (20.9)	<0.001 ‡
Three	3 (1.9)	6 (3.3)	0.513 ‡

Values are presented as mean ± standard deviation or number (%). * *p*-value obtained by student *t*-test, † *p*-value obtained by chi-square test, ‡ *p*-value obtained by Fisher’s exact test. *p*-value of less than 0.05 was considered statistically significant. BMI: Body mass index, IUD: Intrauterine device, MHT: Menopause hormonal therapy.

**Table 2 medicina-57-00577-t002:** Distribution of various pathogens and other microorganisms as identified using both tests.

Variable	Total (%)	Group A (*n* = 182)	Group B (*n* = 157)	*p*-Value
**STD12-Multplex real-time PCR test**				
*Gadnerella vaginalis*	220 (64.9)	104 (66.2)	116 (63.7)	0.630 †
*Candida albicans*	60 (17.7)	31 (19.7)	29 (15.9)	0.326 †
*Myocoplasma hominis*	35 (10.3)	14 (8.9)	21 (11.5)	0.429 †
*Mycoplasma genitalium*	1 (0.3)	0 (0)	1 (0.5)	1.000 ‡
*Ureaplasma parvum*	130 (38.3)	81 (51.6)	49 (26.9)	<0.001 †
*Ureaplasma urealyticulum*	36 (10.6)	23 (14.6)	13 (7.1)	0.025 †
*Chlamydia trachomatis*	3 (0.9)	1 (0.6)	2 (1.1)	1.000 ‡
*Trichomonas vaginalis*	4 (1.2)	4 (2.5)	0 (0)	0.045 ‡
Herpes simplex virus type 2	14 (4.1)	8 (5.1)	6 (3.3)	0.407 †
**Routine culture**				
Aerobic bacterial microoranism	152	62 (39.5%)	90 (49.4%)	0.066 †
*Escherichia coli*	54 (15.9)	22 (14.0)	32 (17.6)	0.370 †
*Streptococcus agalactiae*	36 (10.6)	20 (12.7)	16 (8.8)	0.239 †
Streptococcus anginosus group	7 (2.1)	2 (1.3)	5 (2.7)	0.457 ‡
*Streptococcus mitis*	1 (0.3)	0 (0)	1 (0.5)	1.000 ‡
*Staphylococcus aureus*	11 (3.2)	2 (1.2)	9 (4.9)	0.023 ‡
*Enterococcus faecalis*	18 (5.3)	10 (6.4)	8 (4.4)	0.419 †
Corynebacterium Group	5 (1.5)	0 (0)	5 (2.7)	0.064 ‡
*Enterobacter aerogenes*	4 (1.2)	1 (0.6)	3 (1.6)	0.627 ‡
*Klebsiella pneumoniae*	10 (2.9)	2 (1.3)	8 (4.4)	0.114 ‡
*Pseudomonans aeruginosa*	2 (0.6)	0 (0)	2 (1.1)	0.501 ‡
Others	4 (1.2)	3 (1.9)	1 (0.5)	0.340 ‡
*Candida* spp.				
*Candida albicans*	60 (17.7)	31 (19.7)	29 (15.9)	0.326 ‡
Non-albicans Candida	12 (3.5)	7 (4.5)	5 (2.7)	0.395 †
**Total patients of Cadida species**	70 (20.6)	38 (24.2)	32 (17.5)	0.133 †
Sexually transmitted disease	21 (6.2)	12 (3.5)	9 (2.7)	0.304 †

† *p*-value obtained by chi-square test, ‡ *p*-value obtained by Fisher’s exact test. *p*-value of less than 0.05 was considered statistically significant. PCR: polymerase Chain Reaction, Total patients of Candida species: total patients of *Candida* spp. in both tests.

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
