# Peer review of "Prevalence of Pathogens and Other Microorganisms in Premenopausal and Postmenopausal Women with Vulvovaginal Symptoms: A Retrospective Study in a Single Institute in South Korea"

_medicina, 2021, doi:10.3390/medicina57060577_

Round 1

Reviewer 1 Report

Dear authors

There was some improvement, but you did not improve the discussion - it needs extensive improvement before being reviewed. I pointed a few issues, but there are many more.

Line 34 – you should refer to presence of Candida (you are talking about the presence of microorganisms)

- Line 96-7 – unusual menopause criteria

- Clarify:   since it was a retrospective study, I assume that at your institution you systematically perform cultures and PCR for all women – is that correct? Wet mount or Nugent performed? If all this is performed by routine, overtreatment is very likely! Please comment in the manuscript

- No rationale in testing HSV2 and not HSV1 (the rate of genital isolations of HSV1 has been increasing!)

- Line 174 – do not assume infection – it can be colonization

- Line 180 – same as before – the presence of E. coli is not necessarily an “infection”

- In table 2 do not think renaming AV as “bacterial microorganisms” will work – i.e. Gardnerella are also bacteria. Also, delete the rows with the number of “pathogens”

- Line 224 – and other microorganisms

- Line 231 – you can only say it was present – not necessarily “vaginitis” agent; BV is much more complex than just identifying Gardnerella

- Line 239-40 – of note, wet mount works almost as well

- Line 249 – more tests available: i.e.10.1111/1471-0528.16661

- The discussion on the molecular diagnosis of BV needs to be improved

- Line 258 – it is not

- Line 261 – it does not!! That’s the opposite: less Candida and less BV

- Line 266-7 – please review – it is not true. BV is much lower in postmenopausal women, even after correction with estrogens

- Line 273 - “Vulvovaginal candidiasis (VVC) was the third most common species” – VVC is a disease, not a species (and when you talk about Candida, its genera)

- The discussion is still too long and often unrelated to the results.

- Please review the results and rewrite de discussion accordingly. Keep in mind that in your study you merely tested the presence of microorganism (some of which of very doubtful interest). You cannot assume great conclusions concerning vaginitis, since you did not compare your findings with the gold standard.

- One important message is that what you have done should not be used in clinical practice, as it leads to overtreatments!

Author Response

Dear Trista Tang

 Thank you for considering our manuscript (medicina-1240169). We also thank the reviewers for their helpful comments on the paper. We have enclosed the revised manuscript with a list of changes, which are described in the following point-by-point replies to the reviewers’ comments.

=============================================================

Point-by-point replies to the comments by Reviewer #1

  1. Line 34 – you should refer to presence of Candida (you are talking about the presence of microorganisms)

We have changed this sentence as follows:

  • Before: Ureaplasma parvum was the second most common microorganism, followed by vulvovaginal candidiasis. (line 34 of the original version)

  • After: Ureaplasma parvum was the second most common microorganism, followed by Candida species.

  1.  unusual menopause criteria

We have changed this sentence as follows:

- Before: Postmenopausal state was defined as amenorrhea for more than 6 months in women aged 50 years or more, or the diagnosis of menopause with a Follicle Stimulating Hormone (FSH) level of 40 mIU/mL or more in women aged 50 years or younger.

- After: Postmenopausal state was defined as amenorrhea for more than 12 months in women aged 50 years or more, or the diagnosis of menopause with a Follicle Stimulating Hormone (FSH) level of 25 mIU/mL or more in women aged 50 years or younger.

  1. Clarify:   since it was a retrospective study, I assume that at your institution you systematically perform cultures and PCR for all women – is that correct? Wet mount or Nugent performed? If all this is performed by routine, overtreatment is very likely! Please comment in the manuscript

For all patients with symptoms of vaginitis or suspected of vaginitis, we performed culture, STD, and Gram staining. Wet smear was performed if only necessary. As a retorspective study, we want to present the prevalence of microorganisms through results of both tests. I think that it is unreasonable to discuss about overtreament in this manuscript.

  1.  No rationale in testing HSV2 and not HSV1 (the rate of genital isolations of HSV1 has been increasing!)

We agree with this comment. The current use of our PCR test does not include HSV1, and this paper does not study all microorganisms that cause vulvovaginal symptoms. Testing for HSV1 in all patients will also be an overtreatment.

  1. Line 174 – do not assume infection – it can be colonization

=> We agree with this comment. However, the diagnosis can be made in a woman who has signs and symptoms of vaginitis when either 1) a wet preparation (saline, 10% KOH) or Gram stain of vaginal discharge demonstrates budding yeasts, hyphae, or pseudohyphae or 2) a culture or other test yields a positive result for a yeast species. (2015 Sexually transmitted diseases treatment guideline, CDC – part of vulvovaginal candidiasis)

We have corrected the error (Table 2)

Vulvovagianl candidiasis -> total number of Cadida species.

- Befor: There was no statistically significant difference in either group. In this study, the prevalence of VVC was 70 (20.6%).

- After: There was no statistically significant difference in either group. In this study, the prevalence of Candida species was 70 (20.6%)

  1.  Line 180 – same as before – the presence of E. coli is not necessarily an “infection”

We have changed this sentence as follows:

- Before: In single infection with aerobic bacteria group, the most common bacteria were E. coli (n=15), followed by GBS (n=7), Enterococcus faecalis (E. faecalis) (n=6), Klibsiella pneumoniae (K. pneumonia) (n=2), Staphylococcus aureus (S. aureus) (n=2) and Pseudomonas aeruginosa (P. aeruginosa) (n=1).

- After: In single detection with aerobic bacterial group, the most common bacteria were E. coli (n=15), followed by GBS (n=7), Enterococcus faecalis (E. faecalis) (n=6), Klibsiella pneumoniae (K. pneumonia) (n=2), Staphylococcus aureus (S. aureus) (n=2) and Pseudomonas aeruginosa (P. aeruginosa) (n=1).

  1.  In table 2 do not think renaming AV as “bacterial microorganisms” will work – i.e. Gardnerella are also bacteria. Also, delete the rows with the number of “pathogens”

We have corrected the error (Table 2)

(Aerobic vaginitis -> Aerobic bacterial microorganism)

  1. Line 224 – and other microorganisms

We have changed this sentence as follows:

   - Before: This study investigated the prevalence of pathogens and microorganisms in premenopausal and postmenopausal women with vulvovaginal symptoms using STD 12-Multiplex real-time PCR and routine culture of vaginal specimens.

   - After: This study investigated the prevalence of pathogens and other microorganisms in premenopausal and postmenopausal women with vulvovaginal symptoms using STD 12-Multiplex real-time PCR and routine culture of vaginal specimens.

  1. Line 231 – you can only say it was present – not necessarily “vaginitis” agent; BV is much more complex than just identifying Gardnerella

We agree with this comment:

  1.  Line 239-40 – of note, wet mount works almost as well

We agree with this comment:

  1. Line 249 – more tests available: i.e.10.1111/1471-0528.16661

We agree with this comment, we have changed this sentence as follows:

- Before: The other results of real-time multiplex PCR assay for bacterial vagionsis have a sensitivity of 90.5%, a specificity of 85.8%, a positive predictive value of 89%, and a negative predictive value of 87.7%, compared to Nugent score and/or Amsel’s criteria

- After: The recent results of real-time multiplex PCR assay for bacterial vagionsis have a sensitivity of 91.7% and a specificity of 86.6%, compared to Nugent score and/or Amsel’s criteria.

  1.  Line 258 – it is not

We agree with this comment, we have deleted that sentence:

  1. vaginalis is not only a direct cause, but the biofilm produced by the organism also plays an important role in the occurrence of vaginitis. Biofilms are mucous-like materials made by bacteria. The bacteria can hide under the biofilms, thus making antibiotics ineffective. In the vagina, these biofilms are mainly composed of G. vaginalis and sometimes Atopobium vaginae[11]. Hence, it is assumed that G. vaginalis plays an important role in the diagnosis and management of suspected as well as symptomatic vaginitis.

  1. Line 261 – it does not!! That’s the opposite: less Candida and less BV.
  2. - Line 266-7 – please review – it is not true. BV is much lower in postmenopausal women, even after correction with estrogens.

We agree with this comment, we have changed this sentence as follows:

- Before: Changes in the vaginal pH, vaginal microflora, and cellular glycogen content occur due to endocrine changes caused by menopause[14]. This alkaline vaginal condition (pH>4.5) due to aging leads to a higher occurrence of bacterial vaginosis in menopausal women[15]. In addition, asymptomatic carriers of G. vaginalis and gram-negative bacteria are more common among postmenopausal women[16, 17]. These factors play an important role in the development of vaginal infection by inducing an imbalance in the vaginal microflora. Being sexually active is one of the many risk factors associated with bacterial vaginosis, and menopause might also be included in this group.

- After: Changes in the vaginal pH, vaginal microflora, and cellular glycogen content occur due to endocrine changes caused by menopause[14]. Menopausal women showed a more alkaline vaginal condition (pH>4.5)[15]. In addition, asymptomatic carriers of G. vaginalis and gram-negative bacteria are more common among postmenopausal women[16, 17]. These factors play an important role in the development of vaginal infection by inducing an imbalance in the vaginal microflora. Being sexually active is one of the many risk factors associated with vaginitis such as bacterial vaginosis, and menopause might also be included in this group.

  1. Line 273 - “Vulvovaginal candidiasis (VVC) was the third most common species” – VVC is a disease, not a species (and when you talk about Candida, its genera)

We agree with this comment, we have changed this sentence as follows:

   -Before: Vulvovaginal candidiasis (VVC) was the third most common species identified in this study, after bacterial vaginosis caused by G.vaginalis and U. parvum.

   - After: Candida species was the third most common those identified in this study, after bacterial vaginosis caused by G.vaginalis and U. parvum.

Reviewer 2 Report

It is common practice in many European countries to empirical treatment of women with vaginitis on the results of gynecological examination and/or reported complaints. Doing so can transform acute vaginitis into chronic inflammation with known pathogen-dependent serious sequelae. In addition, a woman's sexual partner is often overlooked in the treatment process. Failure to treat a sick woman's partner is a natural cause of disease recurrence. It is especially dangerous not to recognize vaginitis caused by certain pathogens in women during the procreation period. In special cases, the consequence of such omission may be secondary infertility or serious consequences in newborn children.

The purpose of the submitted study was to determine the causes and prevalence of vaginitis in premenopausal and postmenopausal women, using STD 12-Multiplex real-time PCR test and routine culture of vaginal discharge.

To achieve the assumed goal, the authors used appropriate laboratory tools and based the analysis of the research material on well-selected statistical methods.

The authors showed in the study that the prevalence of bacterial vaginosis on PCR was 64.9%, which is higher than that reported in other studies. In addition, the previous study reported that sexually active of women had a higher prevalence of bacterial vaginosis than the menopausal group, which is different from the results of this study. In a substantive discussion, the authors explain the mentioned differences between their own results and literature data.

The discussion on the importance of ureoplasma and mycoplasma infections in the pathogenesis of vaginitis in both reproductive and postmenopausal women is of particular interest. The discussion culminates in critical remarks with regard to drawing unequivocal conclusions from one's own research, taking into account, inter alia, various research methods, institutional references, and the retrospective nature of the study. The final conclusion of the study is justified and confirms my opinion that the correct selection of diagnostic methods is crucial for the effective diagnosis and treatment of inflammation of the lower genital tract in women.

Author Response

Dear Reviewer

 Thank you for your consideration of our manuscript (medicina-1240169). We also thank the reviewers for their helpful comments on the paper. We are enclosing the revised manuscript with a list of changes.

=============================================================

Thank you for comments and suggestions about my manuscript.

This manuscript is a resubmission of an earlier submission. The following is a list of the peer review reports and author responses from that submission.

Round 1

Reviewer 1 Report

Abstract

- Infectious, symptomatic “vaginitis” are not more common in the menopause, contrarily to what is suggest in the background

- There is some confusion between “vaginitis” and STIs

- Do not think you can assume that you tested for “vaginitis” – a positive culture or PCR for Gardnerella cannot be assumed as BV; also, culture or identification of Candida does not distinguish infection from colonization. Most women, even if asymptomatic, will harbor Gardnerella spp. Comparing, for instance, prevalence of GBS is not relevant: it is more common in women with DIV but can also be cultured from women with normal flora.

- Most relevant fact (and possibly the only one that actually can be interpreted) is the difference in T. vaginalis. Give more details about that.

- The conclusions do not derive from the results

- One of the weak points of the study is that no comparison with the gold standard for BV was performed. Also, other causes such as DIV and cytolytic vaginosis were not considered.

- Line 42 – not all “vaginitis” are inflammatory: BV, cytolytic vaginosis, vaginal atrophy (≠atrophic vaginitis)

- Line 44 – your data refers to South Korea – not relevant to show US data (a broader worldwide vision would be more useful or, preferably, from SK). Most cases of BV are asymptomatic.

- Line 49 – highlight that Gram stain is usually Nugent score

- Line 54-56 – confusing – while it is true, it decrease the risk of Candida, BV, cytolytic vaginosis

- Line 67-69 – since it was a retrospective study, I assume that at your institution you systematically perform cultures and PCR for all women – is that correct? Wet mount or Nugent performed?

- Line 70 – give an idea of the number of patients evaluated at your department during that period

- From where exactly were the samples collected?

- Line 82-85 – that would definitely be my choice of microorganisms to test – the lack of non-albicans Candida is clearly a weak point. Why HSV2 and not 1 (HSV1 is more and more common in genital herpes!). Many of these agents may be mere colonizers. Dangerous to mix STIs and often innocent bacteria, such as Gardnerella. These tests can easily lead to overtreatments! (on the strong points, can detect unsuspected true and dangerous STIs). I suppose you are using the test as part of an experience

- Line 99-101 – do you mean by culture? Some need cellular culture

- Ethical approval granted?

- Bleeding, vaginal medication, recent antibiotics as exclusion criteria?

- Exclusion of other vulvovaginal diseases? How was vaginal atrophy/atrophic vaginitis excluded as the cause for symptoms?

- Evaluation for dermatosis or vulvodynia?

Results

- It is surprising how the groups were so similar in terms of numbers – would expect more in the premenopausal group

- For the reader it probably would be easier to follow if group A were the premenopausal

- Number of symptoms of vaginitis not a validated criterion (table)

- Table 2 – cannot assume the presence of a microorganism as the cause of the symptoms.

How did you define aerobic vaginitis? Only based on cultures? The diagnosis is defined by microscopy, despite some attempts to do it resorting to PCR

Table 3 – AV as a single infection??; eliminate table

Line 133 – not always a pathogen (in some series, cultivable from >60% of women); cannot assume UP as causative of “vaginitis” symptoms (…)

- Line 137/8 – unaware of the criteria for the diagnosis of AV by culture

- 142/3 - ?

- Line 162/177 – delete

- Line 178 – microorganisms more accurate than pathogens

Line 186 – the study did not evaluate the prevalence nor the cause of vaginitis – it evaluated the presence of microorganisms (some pathogenic, others not necessarily) in women with vulvovaginal symptoms

- Line 193-4 – No!!

- Line 195-6 – the presence of Gardnerella is not synonymous of BV. The molecular diagnosis of BV is possible, but includes evaluating other BVAV and lactobacilli

- Line 201-2 – very poor performance

- Reference 10 is from 2010 – please check more recent references on the molecular diagnosis of BV (BD, Aptima, Seegene)

The discussion is very long and often unrelated to the results.

Please review the results and rewrite de discussion accordingly

Reviewer 2 Report

It is common practice in many European countries to empirical treatment of women with vaginitis on the results of gynecological examination and/or reported complaints. Doing so can transform acute vaginitis into chronic inflammation with known pathogen-dependent serious sequelae. In addition, a woman's sexual partner is often overlooked in the treatment process. Failure to treat a sick woman's partner is a natural cause of disease recurrence. It is especially dangerous not to recognize vaginitis caused by certain pathogens in women during the procreation period. In special cases, the consequence of such omission may be secondary infertility or serious consequences in newborn children.

The purpose of the submitted study was to determine the causes and prevalence of vaginitis in premenopausal and postmenopausal women, using STD 12-Multiplex real-time PCR test and routine culture of vaginal discharge.

To achieve the assumed goal, the authors used appropriate laboratory tools and based the analysis of the research material on well-selected statistical methods.

The authors showed in the study that the prevalence of bacterial vaginosis on PCR was 64.9%, which is higher than that reported in other studies. In addition, the previous study reported that sexually active of women had a higher prevalence of bacterial vaginosis than the menopausal group, which is different from the results of this study. In a substantive discussion, the authors explain the mentioned differences between their own results and literature data.

The discussion on the importance of ureoplasma and mycoplasma infections in the pathogenesis of vaginitis in both reproductive and postmenopausal women is of particular interest. The discussion culminates in critical remarks with regard to drawing unequivocal conclusions from one's own research, taking into account, inter alia, various research methods, institutional references, and the retrospective nature of the study. The final conclusion of the study is justified and confirms my opinion that the correct selection of diagnostic methods is crucial for the effective diagnosis and treatment of inflammation of the lower genital tract in women.